# Diagnosis and Treatment for Gastric Mucosa-Associated Lymphoid Tissue (MALT) Lymphoma

**DOI:** 10.3390/jcm12010120

**Published:** 2022-12-23

**Authors:** Shotaro Nakamura, Mariko Hojo

**Affiliations:** 1Department of Gastroenterology, International University of Health and Welfare, Narita 286-8686, Japan; 2Center of Gastroenterology, Takagi Hospital, Fukuoka 831-0016, Japan; 3Department of Gastroenterology, Juntendo University School of Medicine, Tokyo 113-8421, Japan

**Keywords:** gastric lymphoma, MALT lymphoma, *Helicobacter pylori*, radiotherapy, chemotherapy, rituximab

## Abstract

Mucosa-associated lymphoid tissue (MALT) lymphoma, which was first reported in 1984, shows an indolent clinical course. However, the detailed clinicopathological characteristics of gastric MALT lymphoma have not been fully elucidated. We performed a literature search concerning the clinical features and treatment for gastric MALT lymphoma using PubMED. MALT lymphomas develop in single or multiple extranodal organs, of which the stomach is one of the most frequent sites; gastric MALT lymphoma accounts for 7% to 9% of all B-cell lymphomas, and 40% to 50% of primary gastric lymphomas. The eradication of *Helicobacter pylori* (*H. pylori*) is the first-line treatment for patients with gastric MALT lymphoma, regardless of the clinical stage. Approximately 60–90% of cases with stage I/II_1_ disease only achieve a complete histological response via *H. pylori* eradication. In patients who do not respond to *H. pylori* eradication therapy, second-line treatments such as watch-and-wait, radiotherapy, chemotherapy, rituximab immunotherapy, and/or a combination of these are recommended. Thus, *H. pylori* plays a causative role in the pathogenesis of gastric MALT lymphoma, and *H. pylori* eradication leads to complete histological remission in the majority of cases.

## 1. Introduction

Extranodal marginal zone lymphoma of mucosa-associated lymphoid tissue (MALT lymphoma) was first described by Isaacson and Wright in 1984 [1]. They reported four cases of MALT lymphoma, with one case each presenting in the stomach, salivary gland, lung, and thyroid. MALT lymphoma is a low-grade non-Hodgkin’s lymphoma composed of small- to medium-sized neoplastic B cells including marginal-zone centrocyte-like cells, monocytoid cells, and scattered immunoblasts [2,3,4,5,6,7,8,9]. MALT lymphoma most frequently develops within the stomach. In most cases of gastric MALT lymphoma, *Helicobacter pylori* (*H. pylori*) is involved in the pathogenesis, and the eradication of *H. pylori* results in the complete histological response of the lymphoma in 60–90% of patients [2,3,4,5,6,7,8,9]. In the present article, the authors review the recent findings concerning the pathogenesis, diagnosis, and treatment strategies for cases of gastric MALT lymphoma.

## 2. Epidemiology of Gastric MALT Lymphoma

Primary gastric lymphoma (PGL) is the most frequent extranodal non-Hodgkin’s lymphoma, comprising 30–40% of all extranodal lymphomas. Among the 676 cases of primary gastrointestinal lymphoma treated at our institution [10,11], the most commonly affected site was the stomach (456 cases (67%)), followed by the small and large bowels (183 cases; (27%)). The remaining 37 cases (5%) had lymphomas in both the stomach and intestines. The most frequent histologic type of gastric lymphoma was MALT lymphoma (226 cases (51%)), followed by diffuse large B-cell lymphoma (DLBCL) (170 cases (39%)).

Gastric MALT lymphoma comprises 7–9% of all B-cell lymphomas, 3–6% of all malignant neoplasms in the stomach, and 40–50% of all PGLs [2,3,4,5,10,11,12,13,14]. While gastric MALT lymphoma is relatively rare, its incidence has been increasing over the past 20–30 years. In a nationwide survey using the Dutch nationwide histopathology registry (PALGA), Capelle et al. identified 1419 newly diagnosed cases of gastric MALT lymphoma over a 16-year period [15]. In addition, they and other authors reported an increased risk for the metachronous development of gastric adenocarcinoma after gastric MALT lymphoma [15,16].

## 3. Pathogenesis of Gastric MALT Lymphoma

### 3.1. Helicobacter pylori

The etiological association of *H. pylori* with gastric MALT lymphoma was first reported by Wotherspoon et al. in 1991 [4]. Thereafter, several clinical and histopathologic studies showing a definite association between *H. pylori* infection and gastric MALT lymphoma were confirmed [1,2,3,4,5,6,7].

Approximately 80–90% of patients with gastric MALT lymphoma are infected with *H. pylori*, and 60–80% of those cases achieve remission via *H. pylori* eradication alone [5,6,7,8,9]. In *H. pylori*-dependent cases, the growth of lymphoma cells is driven by *H. pylori*-generated immune responses that include signaling from CD40 and CD86 through the aid of bystander T-cells [2,3]. In addition, a proliferation-inducing ligand (APRIL), which is a member of the tumor necrosis factor (TNF) superfamily and is produced by eosinophils in *H. pylori*-infected gastric mucosa, promotes the survival and proliferation of neoplastic B cells [17,18]. Interestingly, *H. pylori* eradication therapy is not only effective in *H. pylori*-positive cases but also in *H. pylori*-negative cases. In our experience, the complete remission (CR) rate of MALT lymphoma after *H. pylori* eradication therapy was 75% among *H. pylori*-positive cases, while it was 29% among *H. pylori*-negative cases [13,14].

It should be noted that t(11;18)(q21;q21)/*BIRC3-MALT1* translocation is frequent (53% or 60%) in *H. pylori*-negative gastric MALT lymphomas [19,20,21,22]. Recently, Tanaka et al. obtained gastric mucosal biopsy specimens from *H. pylori*-negative gastric MALT lymphoma patients and from control participants without *H. pylori* infection or cancer and studied their mucosal microbiota by performing 16S rRNA gene sequencing. They found that the *H. pylori*-negative gastric MALT lymphoma patients showed lower alpha diversity than the control participants (*p* = 0.04) [23]. A comparison concerning the patients’ beta diversity revealed that there were significant differences in the microbes that were present in the gastric mucosa between the two groups (*p* = 0.04). They concluded that *H. pylori*-negative gastric MALT lymphoma cases exhibited altered gastric mucosal microbial compositions, and such altered microbiota might be involved in the pathogenesis of *H. pylori*-negative gastric MALT lymphoma [23].

In addition, patients with gastric MALT lymphoma have an increased risk of the metachronous development of gastric adenocarcinoma [15,16]. Palmela et al. reported that the relative risk (RR) for gastric adenocarcinoma among patients who had gastric MALT lymphoma was 4.32 (95% CI 2.64–6.67) compared to the American population. The median latency time was 5 years, and the risk was maintained thereafter [RR 4.92, 95% confidence interval (CI) 2.45–8.79] [16].

### 3.2. Genetic Aberrations

Genetically, MALT lymphoma is associated with the following four chromosomal translocations: t(11;18)(q21;q21)/*BIRC3-MALT1*, t(1;14)(q22;q32)/*BCL10*-*IGH*, t(14;18)(q32;q21)/*IGH*-*MALT1*, and t(3;14)(q31;q32)/*FOXP1*-*IGH* [2,3,11,19,20,21,22]. Especially in MALT lymphomas with t(11;18)(q21;q21)/*BIRC3-MALT1*, the resulting BIRC3-MALT1 fusion product forms self-oligomers via a non-homotypic interaction mediated by the BIRC3 moiety, leading to constitutive nuclear factor kappa B (NF-κB) activation. These oncogenic products are most likely synergic with both innate and acquired immune stimulations with respect to the activation of the NF-κB pathway [18,19,20,21,22,23,24,25,26].

All four above-mentioned translocations are considered to exert their oncogenic activities alongside the constitutive activation of the NF-κB pathway, which leads to the expression of several genes for cell survival and proliferation [24,25,26,27,28]. Among these translocations, t(11;18)(q21;q21)/*BIRC3-MALT1* is the most frequent, and is detected in 15–30% of all cases of gastric MALT lymphomas. The translocation fuses the N-terminal region of *BIRC3* to the C-terminal region of *MALT1*, generating a functional chimeric fusion protein that gains the ability to activate the NF-κB pathway [25,26,27]. The t(11;18) (q21;q21)/*BIRC3-MALT1* translocation is frequently associated with the absence of *H. pylori*, and cases that are positive for this translocation do not respond to *H. pylori* eradication [19,20,21,22]. Interestingly, t(11;18)(q21;q21)-positive MALT lymphomas occasionally transform into DLBCL [20,21].

The TNF-α-induced protein 3 gene (*TNFAIP3*, *A20*), which was identified as the target of 6q23 deletion in MALT lymphoma, is an important inhibitor of NF-κB [27]. Mutations or deletions of *A20* that lead to A20 protein inactivation are frequent in MALT lymphomas of the ocular adnexa, salivary glands, thyroid, and liver. A20-mediated oncogenic activities in MALT lymphoma depend on NF-κB activation triggered by TNF-α or other unidentified molecules [25,26,27].

## 4. Diagnosis of Gastric MALT Lymphoma

### 4.1. Endoscopic/Macroscopic Findings

To date, a standard endoscopic or macroscopic classification system for PGLs has not been established. In Western nations, PGLs are macroscopically classified as either ulcerative (34–69%), mass/polypoid (26–35%), diffuse-infiltrating (15–40%), or other types [6,7,28]. In our institute, PGLs have been classified as either superficial (41%), mass-forming (43%), diffuse-infiltrating (6%), or other types (10%). Figure 1 shows conventional and magnified endoscopic findings of superficial-type gastric MALT lymphomas. In Figure 1, the upper three photos show conventional endoscopic images, while the lower three photos are magnified endoscopic images that show a characteristic tree-like appearance (TLA) with amorphous, whitish areas. Superficial-type gastric MALT lymphoma is sometimes misdiagnosed as depressed-type early gastric cancer [6,11]. In an article by Lee et al., endoscopic findings of 122 cases of gastric MALT lymphoma were classified as either polypoid (*n* = 18, 15%), ulceration (*n* = 43, 35%), or diffuse infiltration (*n* = 61, 50%) types [28].

### 4.2. Histopathology of Gastric MALT Lymphomas

A definite diagnosis of gastric MALT lymphoma is made based on the histopathologic criteria of the World Health Organization (WHO) classification system [2,3], the Consensus report of the European Gastro-Intestinal Lymphoma Study (EGILS) group [29], and the National Comprehensive Cancer Network (NCCN) guidelines [30]. Histologically, the diffuse infiltrate of atypical neoplastic lymphoid cells (centrocyte-like cells) around reactive follicles showing a marginal-zone growth pattern, which often infiltrate into gastric glands causing the destruction of epithelial cells (lymphoepithelial lesions (LELs), Figure 2), is observed in gastric MALT lymphomas [1,2,3,4]. MALT lymphoma cells immunohistochemically exhibit CD20+, CD79a+, CD5-, CD10-, CD23-, CD43+/-, and cyclin D1-. However, when large lymphoma cells are present in a solid or sheet pattern, a diagnosis of DLBCL should be made [6,7,10,11]. DLBCL usually shows high Ki-67 expression upon immunohistochemical staining [6,19,20]. The histologic diagnosis of gastric lymphomas should be confirmed by an expert hematopathologist [1,2,3].

The histological features of MALT lymphoma are occasionally similar to those of reactive inflammatory conditions such as *H. pylori*-related chronic gastritis. Histologically, MALT lymphoma can be distinguished from gastritis based on the presence of a dense infiltrate of monotonous B-cells extending away from lymphoid follicles, the cytologic atypia of lymphoid cells, Dutcher bodies, and LELs in MALT lymphomas [2,3]. The Wotherspoon score is used to make a confident histologic diagnosis of gastric MALT lymphoma on examination of biopsy specimens (Table 1) [4].

For the diagnosis of gastrointestinal MALT lymphoma, a minimum of 10 biopsy samples should be taken, and not only from visible lesions, but also from endoscopically normal-appearing mucosa, according to the Consensus report of the EGILS group [29]. The detection of immunoglobulin light chain (κ or λ) restriction by immunohistochemistry or in situ hybridization, and analyses for clonality of the rearranged immunoglobulin genes by polymerase chain reaction (PCR), may help diagnose B-cell lymphoma [2,3]. Cytogenetic analyses using G-banding, reverse-transcription polymerase chain reaction (RT-PCR), and/or fluorescence in situ hybridization (FISH) to detect t(11;18) (q21;q21)/*BIRC3-MALT1* and/or other specific chromosomal translocations can be performed to confirm the certain diagnosis of MALT lymphoma [2,3,19,20,21,22].

### 4.3. Clinical Staging

Clinical staging is essential for determining the management strategy for PGLs. In general, the Lugano International Conference (Blackledge) classification system (I, II_1_, II_2_, IIE, or IV, Table 2) [30] and/or the Ann-Arbor staging system with its modifications by Musschoff and Radaszkiewicz (I_1_E, I_2_E, II_1_E, II_2_E, IIIE, or IV) are used for patients with gastrointestinal lymphomas. Recently, the EGILS Consensus report and the European Society of Medical Oncology (ESMO) guidelines recommended the Paris Classification system, which is a modification of the TNM system including the degree of the spread of lymphoma as assessed by endoscopic ultrasound (EUS) [29]. In our institute, we have combined the Lugano system and Paris system, as shown in our original Table 2. Esophagogastroduodenoscopy (EGD) with or without multiple biopsies, ileocolonoscopy, and balloon-assisted enteroscopy (BAE) are useful for detecting possible gastrointestinal lesions of lymphomas during a follow-up. In addition, computed tomography (CT) of the chest, abdomen, and pelvis, as well as fluorine-18 [^18^F] fluoro-deoxyglucose positron-emission tomography (FDG-PET), should be considered [31].

Cohen et al. performed an analysis of 66 patients with MALT lymphoma to assess the usefulness of FDG-PET. They found that all extranodal lesions of MALT lymphoma located in subcutaneous tissues, breasts, lungs, and liver were detected by FDG-PET [31]. However, only 26.3% of the lesions located in the stomach and 28.6% of the lesions located in the intestinal tract were detected by FDG-PET. They evaluated the predictive value of PET in the patients and found that increased ^18^F-FDG-uptake in extranodal lesions was associated with disease progression [31]. The gastrointestinal tract has high/heterogeneous physiologic ^18^F-FDG-uptake. They concluded that the detection rate of extranodal MALT lymphoma lesions located in tissues with low and/or homogenous physiologic [^18^F]FDG-uptake is excellent using [^18^F]FDG-PET-CT.

## 5. Treatments for Gastric MALT Lymphoma

### 5.1. Helicobacter pylori Eradication

The first-line therapy for all patients with gastric MALT lymphoma is *H. pylori* eradication, regardless of the clinical stage [5,6,7,8,9,11,12,13,14,29,30,32,33,34,35,36,37,38,39,40,41,42]. *H. pylori* eradication therapy consisting of a proton-pump inhibitor, the administration of amoxicillin, and either clarithromycin or metronidazole administration for 7 days has been recommended in Japan. The histopathologic evaluation of post-treatment biopsy specimens is conducted based on the Groupe d’Etude des Lymphomes de l’Adulte (GELA) grading system (Table 3 [29]), because the Wotherspoon score (Table 1), which is applied for the initial diagnosis, is no longer adequate for assessing the response to treatment during follow-up [29,42]. Among patients with stage I/II_1_ disease, *H. pylori* eradication therapy resulted in complete histologic response (ChR) in 60–90% of cases [5,7,8,9,10,11,12,13,32,33,35,36,37,38,39,40,41]. However, it should be noted that antibiotic resistance has been the critical factor responsible for treatment failure. In particular, a recent increase in clarithromycin resistance induces treatment failure in *H. pylori* eradication in a considerable number of infected patients.

In a systematic review of 32 published clinical studies including 1408 patients with gastric MALT lymphoma by Zullo et al., the CR rate after *H. pylori* eradication therapy was 78% [33]. The factors associated with resistance to *H. pylori* eradication therapy included the absence of an *H. pylori* infection, an advanced clinical stage, a proximal location of the lymphoma in the stomach, an endoscopic non-superficial type, deep invasion in the gastric wall, and t(11;18)/*BIRC3-MALT1* translocation [11,12,19,20,21,22,25,26,27,28,30,31,33,34,35,36,37,40,41,42,43,44,45]. In addition, Zullo et al. [34] also performed a pooled-data analysis of 315 patients who did not respond to *H. pylori* eradication therapy. The most frequent second-line therapy was radiotherapy (112 patients), followed by surgery (80 patients) and chemotherapy (68 patients). The patients who underwent radiotherapy had a similar remission rate as those who underwent surgical resection (97% vs. 93%; *p* = 0.2) and a significantly higher remission rate than those who underwent chemotherapy (97% vs. 85%; *p* = 0.007).

We conducted a retrospective multicenter study of 420 Japanese patients with gastric MALT lymphoma to assess the long-term clinical outcome after successful *H. pylori* eradication [36]. As a result, CR was achieved in 77% (323/420) of patients solely via *H. pylori* eradication. During the follow-up period (mean 6.5 years and median 6.04 years), treatment failure was seen in only 37 patients (9%) (relapse in 10 of 323 responders; progressive disease (PD) in 27 of 97 non-responders). The long-term prognosis was excellent; the probabilities of freedom from treatment failure, overall survival, and event-free survival after 10 years were 90%, 95%, and 86%, respectively. In addition, we performed another literature review of 32 published articles regarding the efficacy of *H. pylori* eradication in gastric MALT lymphoma cases in our same paper [36], which revealed that *H. pylori* eradication therapy led to CR of gastric MALT lymphoma in 1793 (73%) of 2451 patients.

Subsequently, we performed a prospective multicenter study to evaluate the clinical efficacy of *H. pylori* eradication therapy in 108 Japanese patients with *H. pylori*-infected gastric MALT lymphoma in 34 hospitals (December 2010–February 2016) [45]. As a result, the CR of lymphoma was achieved in 84 (87%) of the 97 patients (11 patients dropped out of the study). Second-line treatments (radiotherapy, rituximab, or gastrectomy) were needed for 10 (10%) of the 97 patients. During the follow-up period (2–45 months; median 5.3 months), three patients died of causes unrelated to lymphoma. The overall survival probability was 97%.

### 5.2. Treatment Strategies for Patients with Gastric MALT Lymphoma Who Do Not Respond to Helicobacter pylori Eradication

There is no consensus on the treatment strategy for patients with gastric MALT lymphoma who do not respond to *H. pylori* eradication therapy. While patients with PD or a clinically evident relapse can undergo oncologic treatment, for patients with persistent histologic lymphoma without PD (responding residual disease (rRD) or no change (NC)), a “watch-and-wait” strategy is recommended for up to 24 months after *H. pylori* eradication therapy. Thereafter, the decision of whether to continue the “watch-and-wait” strategy or to start oncologic treatment is made on a case-by-case basis [11,37].

As for the second-line oncologic treatment, radiotherapy, immunotherapy (e.g., rituximab), and/or chemotherapy are recommended. While radiotherapy and chemotherapy both have a curative potential in localized (stage I/II_1_) gastric MALT lymphoma, radiotherapy (30–40 Gy/15–20 fractions) is generally preferred and is highly effective (response rate of 93–100%) [43,44].

Chemotherapy and/or rituximab immunotherapy are also effective, and these systemic treatments are suitable for cases with histologic transformation into DLBCL, cases with disseminated disease, and those with advanced stages [46,47]. Although rituximab plus cyclophosphamide, doxorubicin, vincristine, and prednisolone (R-CHOP) chemotherapy is relatively toxic for indolent MALT lymphoma, the rituximab plus cyclophosphamide, vincristine, and prednisolone (R-COP) regimen seems to be well-tolerated and effective [45,48,49,50].

Oral alkylating agents, such as cyclophosphamide or chlorambucil, employed as a sole treatment are also well-tolerated and effective, and resulted in CR in 75% of patients, although relapse was observed in 28% of patients [27,41]. Recently, the combination of rituximab and chlorambucil [48], fludarabine [49], and bendamustine [50,51] provided excellent responses in patients with MALT lymphoma of various organs, including gastric MALT lymphoma cases.

In summary, the treatment strategies for patients with gastric MALT lymphoma should be based on the modified version of the ESMO guidelines [29,43,47,51]. For all patients in stages I_1_–IIE, *H. pylori* eradication should be the first-line treatment. In patients with stage IV, *H. pylori* eradication also should be performed if the infection is present.

## 6. Risk for Other Malignancies

It has been reported that patients with gastric MALT lymphoma had a higher risk of synchronous and/or metachronous gastric cancers than healthy control subjects [15,16]. In a Dutch epidemiological study covering the period from 1991–2006 by Capelle et al. [15], new gastric cancers developed in 34 (2.4%) of 1419 patients with gastric MALT lymphoma. The risk of gastric cancer was 16.6 times higher in patients with gastric MALT lymphoma aged 45–59 years than in the Dutch population (*p* < 0.001). They concluded that the risk of gastric cancer in patients with gastric MALT lymphoma is six times higher than that in the Dutch population and recommended that patients with gastric MALT lymphoma should be monitored in thorough follow-ups.

In a US population-based study (National Cancer Institute Surveillance, Epidemiology and End Results 13 (SEER) database 1992–2012), Palmela et al. [16] reported that gastric cancer developed in 20 (0.91%) of 2195 patients with gastric MALT lymphoma, with a relative risk of 4.32 (95% CI 2.64–6.67) compared to the rest of the American population. They concluded that gastric MALT lymphoma is associated with an increased risk of metachronous gastric adenocarcinoma. It should be noted that the risk for gastric cancer was still present beyond 5 years of follow-up [16].

## 7. Discussion

In 1984, Isaacson and Wright reported the first four cases of MALT lymphoma, with one case each arising in the stomach, salivary gland, lung, and thyroid, which was published in the international medical journal *Cancer* [1]. Since then, a large number of clinical studies on the diagnosis and treatment of gastric MALT lymphoma have been published. In 1993, Wotherspoon et al. first reported that *H. pylori* eradication with antibiotic treatment resulted in the complete histological regression of gastric MALT lymphoma in five of six *H. pylori*-infected gastric MALT lymphoma patients [5]. It has been established that *H. pylori* eradication is the first-line treatment for gastric MALT lymphomas. A meta-analysis by Zullo et al. concerning 1408 patients with gastric MALT lymphoma from 32 published studies revealed that after *H. pylori* eradication, 1091 patients (77%) achieved CR, while 43 patients (1.7%) developed progressive disease (PD) and 78 patients (3.1%) experienced a relapse [34]. Thus, treatment failure occurred in only 121 (8.6%) of the 1408 patients.

We performed another meta-analysis of 2451 patients with gastric MALT lymphoma who underwent *H. pylori* eradication therapy from 32 published studies [37]. As a result, 1793 patients (73%) achieved CR after *H. pylori* eradication. Forty-three patients (1.8%) developed PD, while seventy-eight patients (3.2%) suffered a relapse. Thus, only 121 (4.9%) of the 2451 patients experienced treatment failure.

While *H. pylori* eradication therapy is widely accepted as the first-line treatment of gastric MALT lymphoma, the National Comprehensive Cancer Network (NCCN) [30] and ESMO [31,43] guidelines slightly differ regarding which patients should receive antibiotic treatment. NCCN considers *H. pylori* eradication as the first-line therapy of choice for those with early stage lymphoma (Lugano I_1_, I_2_, or II_1_), an *H. pylori*-positive status, and concomitant negativity or unknown status of t(11;18) translocation [30]. Conversely, the ESMO guidelines recommend *H. pylori* eradication therapy for all patients with gastric MALT lymphoma regardless of the stage, translocation status, and *H. pylori* positivity [31,43].

The role of eradication therapy in *H. pylori*-negative MALT lymphoma remains controversial. Since *H. pylori*-negative MALT lymphoma is increasing worldwide, we should discuss it more deeply. Xie et al. [44] performed a meta-analysis regarding the efficacy of modified eradication therapy for *H. pylori*-negative gastric MALT lymphoma concerning 14 studies with 148 patients. As a result, the overall response rate was 0.38 (95% confidence interval (CI): 0.29–0.47). They concluded that their modified *H. pylori* eradication therapy is also effective for patients with *H. pylori*-negative MALT lymphoma.

For patients with gastric MALT lymphoma who do not respond to *H. pylori* eradication therapy and who do not develop PD, a “watch-and-wait” strategy without additional treatment may be adopted. We previously reported that the watch-and-wait strategy resulted in no change in 20% of non-responders with gastric MALT lymphoma after eradication therapy [35]. In addition, more than half of the relapsed cases with gastric MALT lymphoma under the watch-and-wait strategy achieved CR. Thus, we recommend the watch-and-wait strategy for asymptomatic non-responders with early-stage gastric MALT lymphoma. When the clinical stage of patients progresses during the follow-up period, medical treatments including *H. pylori* eradication with or without oncological treatment should be considered. Continuous single-agent chemotherapy with cyclophosphamide or chlorambucil has been useful as a monotherapy [11,41]. However, patients with t(11;18)(q21;q21) translocation may not respond to oral single- or multi-agent immuno/chemotherapy such as rituximab, cladribine, and others.

Non-responders to *H. pylori* eradication therapy have been reported to comprise 17–61% of patients with gastric MALT lymphoma [7,8,9,27,28,48]. As a second-line treatment for such non-responders, radiotherapy (RT) and/or imuno- or chemotherapy should be recommended. Of these, RT at a moderate dose (total 30–36 Gy) is highly effective, although it has been associated with rare acute or late toxicity [51]. Yahalom et al. treated 178 patients with *H. pylori*-independent gastric MALT lymphoma with RT with a median dose of 30 Gy over 20 fractions; consequently, 95% of the patients achieved a complete histologic response [44].

## 8. Conclusions

Nowadays, *H. pylori* eradication therapy is widely accepted as the first-line treatment for patients with gastric MALT lymphoma. To date, a large number of patients with gastric MALT lymphoma have been successfully treated with *H. pylori* eradication therapy. However, there is no consensus regarding the treatment strategies for patients with gastric MALT lymphoma who do not respond to *H. pylori* eradication therapy. In these patients, second-line treatments including a “watch-and-wait” strategy, radiotherapy, chemotherapy, rituximab immunotherapy, or a combination of these should be tailored in consideration of the extent of lymphoma and the clinical stage in each patient.

It is of interest that gastric MALT lymphoma is the only malignancy for which antibiotics are the first choice of treatment with a curative intent [11,41]. Despite recent advances in our understanding of the pathogenesis of gastric MALT lymphoma, there are many unanswered questions. Thus, further clinicopathological and molecular research studies are necessary in order to clarify the detailed mechanisms in the development of gastric MALT lymphoma and the long-term clinical course of patients with said disease.

## Figures and Tables

**Figure 1 jcm-12-00120-f001:**
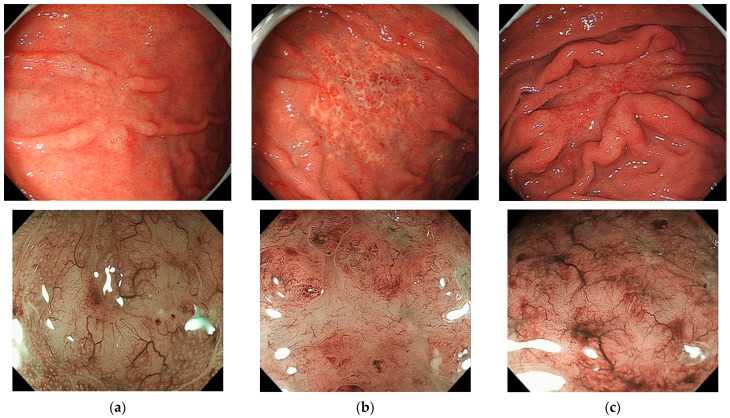
Endoscopic findings of superficial-type gastric MALT lymphomas in three patients (from author’s original research). The upper three photos are conventional endoscopic images and lower three photos are magnified endoscopic images showing a characteristic tree-like appearance with whitish, amorphous areas. (**a**) Patient 1, (**b**) Patient 2, and (**c**) Patient 3.

**Figure 2 jcm-12-00120-f002:**
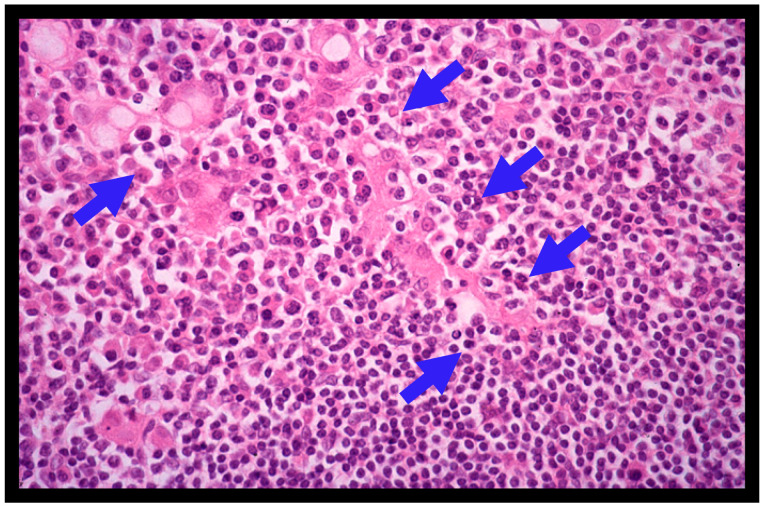
Histological features of gastric MALT lymphoma (H&E; X400). Diffuse infiltrate of small- to medium-sized atypical neoplastic lymphoid cells (centrocyte-like cells) around reactive follicles with a marginal zone growth pattern is shown. Lymphoepithelial lesions (LELs) (blue arrows) are also shown.

**Table 1 jcm-12-00120-t001:** Histologic scoring system for the diagnosis of gastric MALT lymphoma (Wotherspoon score) [4].

Grade	Description	Histologic Features
0	Normal	Scattered plasma cells in mucosa; no lymphoid follicles.
1	Chronic active gastritis	Small clusters of lymphocytes in mucosa; no lymphoid follicles; no lymphoepithelial lesions.
2	Chronic active gastritis with florid lymphoid follicle formation	Prominent lymphoid follicles with surrounding mantle zone and plasma cells; no lymphoepithelial lesions.
3	Suspicious lymphoid infiltrate in mucosa, probably reactive	Lymphoid follicles surrounded by small lymphocytes that have infiltrated diffusely in the mucosa and occasionally into epithelium.
4	Suspicious lymphoid infiltrate in mucosa, probably lymphoma	Lymphoid follicles surrounded by centrocyte-like cells that have infiltrated diffusely in the mucosa and epithelium in small groups.
5	MALT lymphoma	Presence of dense diffuse infiltrate of centrocyte-like cells in the mucosa with prominent lymphoepithelial lesions.

**Table 2 jcm-12-00120-t002:** Lugano and Paris staging systems for gastrointestinal lymphomas [29,30].

Stage	Lugano System	Paris System	Tumor Extension
I	Tumor confined to GI tract (single, primary site or multiple, noncontiguous lesions)	T1m N0 M0	Mucosa
		T1sm N0 M0	Submucosa
		T2 N0 M0	Muscularis propria
		T3 N0 M0	Serosa
II	Tumor extending into abdomen	NA	NA
II_1_	Local nodal involvement	T1–3 N1 M0	Perigastric lymph nodes
II_2_	Distant nodal involvement	T1–3 N2 M0	More distant regional lymph nodes
IIE	Perforation of serosa to involve adjacent organs or tissues	T4 N0–2 M0	Invasion of adjacent structures with or without invasion of abdominal lymph nodes
IV	Disseminated extranodal involvement or concomitant supradiaphragmatic nodal involvement	T1–4 N3 M0	Extra-abdominal lymph nodes
		T1–4 N0–3 M1	and/or additional distant GI/non-GI sites
		T1–4 N0–3 M2	BM not assessed
		T1–4 N0–3 M0–2 BX	BM not involved
		T1–4 N0–3 M2 B1	BM involvement

BM: bone marrow; GI: gastrointestinal; NA: not available.

**Table 3 jcm-12-00120-t003:** GELA histologic grading system for post-treatment evaluation of gastric MALT lymphoma [29].

Score	Lymphoid Infiltrate	LEL	Stromal Changes	Clinical Significance
Complete histologic response (ChR)	Absent or scattered plasma cells and small lymphoid cells in LP	Absent	Normal or empty LP and/or fibrosis	Complete remission
Probable minimal residual disease (pMRD)	Aggregates of lymphoid cells or lymphoid nodules in LP/MM and/or SM	Absent	Empty LP and/or fibrosis	Complete remission
Responding residual disease (rRD)	Dense, diffuse, or nodular, extending around glands in LP	Focal or absent	Focal empty LP and/or absent	Partial remission
No change (NC)	Dense, diffuse, or nodular	Present, may be absent	No changes	Stable or progressive disease

GELA: Groupe d’Etude des Lymphomes de l’Adulte; LEL: lymphoepithelial lesion; LP: lamina propria; MALT: mucosa-associated lymphoid tissue; MM: muscularis mucosae; SM: submucosa.

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
