# Peer review of "Diagnosis and Treatment for Gastric Mucosa-Associated Lymphoid Tissue (MALT) Lymphoma"

_jcm, 2022, doi:10.3390/jcm12010120_

Round 1

Reviewer 1 Report

Almost one third of the references are self citations (this disorder is not so rare and there are a lot of articles discussing it). 

The work is extensive, comprehensive, but I don’t feel it adds novelty to the field. The information is already well known.  Also, the manuscript is a mix between a review and an original article - the authors report literature data, but they also present unpublished data from their own experience, including personal treatment recommendations which do not result from a clinical trial (actually, the research behind this treatment algorithm is not described at all). 

Author Response

Reviewer 1

Open Review

English language and style

( ) English very difficult to understand/incomprehensible
( ) Extensive editing of English language and style required
( ) Moderate English changes required
(x) English language and style are fine/minor spell check required
( ) I don’t feel qualified to judge about the English language and style

Comments and Suggestions for Authors

Almost one third of the references are self citations (this disorder is not so rare and there are a lot of articles discussing it). 

The work is extensive, comprehensive, but I don’t feel it adds novelty to the field. The information is already well known.  Also, the manuscript is a mix between a review and an original article - the authors report literature data, but they also present unpublished data from their own experience, including personal treatment recommendations which do not result from a clinical trial (actually, the research behind this treatment algorithm is not described at all). 

Submission Date

12 November 2022

Date of this review

28 Nov 2022 09:15:18

By way of reply:

  1. Thank you very much for your comments. Our present manuscript is a review article regarding gastric MALT lymphoma, which is a rare, and very interesting disease.
  2. We consider that novelty is not always necessary, since the manuscript is a review article,
  3. It is very interesting that gastric MALT lymphoma is the only malignancy for which antibiotics are the first choice of treatment with curative intent.
  4. Self-citations are only 9 out of all 50 (18%) articles. Another “Nakamura S (Ref. 2 and 3) ” is not me (probably Professor Shigeo Nakamura in Nagoya University).

Reviewer 2 Report

Dr. Nakamura and colleagues reviewed the literature regarding management strategies for patients with gastric MALT lymphoma. They emphasized the central role of H.pylori eradication as a prognostic factor. Besides a thorough review of pathogenesis description and staging elements, they described the therapeutic results of eradicating H.pylori in the histological complete response and remission. They described the alternative management strategies (watch-and-wait, chemotherapy, immunotherapy, combination) in the second line after H.pylori eradication.
The authors should be congratulated for their excellent review of the current literature on this important topic. Here are a couple of comments aiming to improve the manuscript further: 

Major:
- Authors described a standard eradication treatment for H.pylori. However, they might emphasize the growing evidence of antibiotic resistance and the need to enhance the therapy from a « standard » tri therapy (PPI+2 antibiotics) to a quadri therapy vs antibiogram-tailored for resistant strains.

Minor:
- l.239 p. 7/13: wrong reference number. The authors described a study they conducted, however the reference points something not related.

Again I enjoyed reading this important manuscript.

Author Response

Reviewer 2
Open Review
( ) I would not like to sign my review report
(x) I would like to sign my review report
English language and style
( ) English very difficult to understand/incomprehensible
( ) Extensive editing of English language and style required
( ) Moderate English changes required
(x) English language and style are fine/minor spell check required
( ) I don't feel qualified to judge about the English language and style

Comments and Suggestions for Authors
Dr. Nakamura and colleagues reviewed the literature regarding management strategies for patients with gastric MALT lymphoma. They emphasized the central role of H.pylori eradication as a prognostic factor. Besides a thorough review of pathogenesis description and staging elements, they described the therapeutic results of eradicating H.pylori in the histological complete response and remission. They described the alternative management strategies (watch-and-wait, chemotherapy, immunotherapy, combination) in the second line after H.pylori eradication.
The authors should be congratulated for their excellent review of the current literature on this important topic. Here are a couple of comments aiming to improve the manuscript further: Major:
- Authors described a standard eradication treatment for H.pylori. However, they might emphasize the growing evidence of antibiotic resistance and the need to enhance the therapy from a « standard » tri therapy (PPI+2 antibiotics) to a quadruple therapy vs antibiogram-tailored for resistant strains.
Minor:
- l.239 p. 7/13: wrong reference number. The authors described a study they conducted, however the reference points something not related.
Again I enjoyed reading this important manuscript.

Submission Date
12 November 2022 Date of this review
24 Nov 2022 11:49:49

By way of reply:
Antibiotic resistance is the critical factor responsible for eradication treatment failure. Recent increase in clarithromycin resistance induces treatment failure in H. pylori eradication therapy in considerable number of infected patients.

Reviewer 3 Report

Please indicate the authorship or source of figure N°2.

Author Response

Reviewer 3
Please indicate the authorship or source of figure No.2.

By way of reply:
These 6 images are 3 conventional endoscopic images and 3 magnified endoscopic images each in 3 patients with gastric MALT lymphoma treated at Kyushu University Hospital by one of authors (Shotaro Nakamura).

Round 2

Reviewer 1 Report

I do not feel that significant improvements have been made. Also, there is a very high number and self-citations.

Author Response

By way of reply:

  1. Thank you very much for your comments. Our present manuscript is a review article regarding gastric MALT lymphoma, which is a rare, and very interesting disease.
  2. We consider that novelty is not always necessary, since the manuscript is a review article,
  3. It is very interesting that gastric MALT lymphoma is the only malignancy for which antibiotics are the first choice of treatment with curative intent.
  4. Self-citations are only 9 out of all 50 (18%) articles. Another “Nakamura S (Ref. 2 and 3) ” is not me (probably Professor Shigeo Nakamura in Nagoya University).
